# The Role of Potato Glycoside Alkaloids Mediated Oxidative Stress in Inducing Apoptosis of Wolfberry Root Rot Pathogen Fungi

**DOI:** 10.3390/antiox13121537

**Published:** 2024-12-15

**Authors:** Yuyan Sun, Bin Wang, Wei Chen, Yanbo Wang, Dongdong Zhou, Mengyang Zhang, Chongqing Zhang, Ruiyun Li, Jing He

**Affiliations:** 1College of Forestry, Gansu Agricultural University, Lanzhou 730070, China; 13034101180@163.com (Y.S.); wangbin_1519@163.com (B.W.); 18298345663@163.com (W.C.); 15352293633@163.com (Y.W.); ylzd1909@163.com (D.Z.); 18139793655@163.com (M.Z.); zhangchongqing2022@163.com (C.Z.); lruiyun@163.com (R.L.); 2Wolfberry Harmless Cultivation Engineering Research Center of Gansu Province, Lanzhou 730070, China

**Keywords:** wolfberry, root rot, potato glycoside alkaloids, oxidative stress, cell apoptosis

## Abstract

Wolfberry (*Lycium barbarum*) is a vital economic tree species in northwest China, but root rot caused by *Fusarium solani* occurs frequently, which seriously endangers the quality and yield of wolfberry. In this study, potato glycoside alkaloids (PGAs), a plant-derived active substance, were used as materials to explore its inhibitory effect on *F. solani*. By analyzing the changes of reactive oxygen species (ROS) level, antioxidant capacity, and apoptosis, the role of PGAs-mediated oxidative stress in inducing apoptosis of *F. solani* was revealed. The findings suggest that PGAs treatment inhibited mycelium growth, reduced biomass and sporulation, and delayed spore germination in *F. solani*. The concentration for 50% of maximal effect (EC_50_) was 1.85 mg/mL. PGAs treatment induced an increase in caspase-3 activity, disrupting the cell membrane of fungi. In addition, PGAs treatment activated NADH oxidase (NOX) and superoxide dismutase (SOD), promoted hydrogen peroxide (H_2_O_2_) and superoxide anion (O_2_^−^) accumulation, and decreased ascorbate peroxidase (APX), glutathione reductase (GR), and dehydroascorbate reductase (DHAR) activities as well as oxidized glutathione (GSSG), reduced glutathione (GSH), and electron donor NADPH content. In summary, PGAs has a strong inhibitory effect on *F. solani*, and its inhibitory effect may be related to the promotion of ROS accumulation by PGAs, causing the disorder of intracellular redox balance of fungi, the decrease of total antioxidant capacity, and finally the induction of apoptosis. This study provides a new insight into the antifungal mechanism of PGAs against *F. solani*.

## 1. Introduction

Wolfberry (*Lycium barbarum*) is a perennial deciduous shrub that belongs to the *Lycium* genus in the Solanaceae family, and is a pioneer tree species in saline and alkaline areas, widely cultivated in arid and semi-arid areas in northwestern China [1,2]. Wolfberry fruit is considered a superfood due to its unique nutritional and health benefits [3]. However, with the expanding scale of wolfberry cultivation, the occurrence of wolfberry diseases is becoming increasingly serious. Wolfberry root rot is one of the frequent soil-borne diseases of wolfberry plants, often leading to poor plant growth, fruit yield, and quality decline, causing huge economic losses to the wolfberry industry [4]. *Fusarium solani* is one of the main pathogens causing wolfberry root rot, which invades from the root, spreads along the vascular bundle, and eventually leads to plant death [5]. Currently, the control of root rot mainly relies on chemical methods, but the long-term use of chemical fungicides can cause environmental pollution and increased resistance to pathogenic fungi. Thus, it is fundamental to develop new, green, efficient, and broad-spectrum biological control pesticides.

Potato glycoside alkaloids (PGAs) are steroidal secondary metabolites with a wide range of biological activities [6]. The main components of α-solanine and α-chaconine accounted for more than 95% of the total content [7], both of which have inhibitory effects on pathogenic fungi. Studies have demonstrated that α-solanine has a considerable inhibitory effect on the mycelium growth of *Botrytis cinerea* [8], and could also reduce the incidence of carrot soft rot and potato late blight [9]. PGAs treatment not only inhibited the mycelium growth of *Pectobacterium carotovorum* and *F. sulphureum*, but also reduced their pathogenicity to potato [10,11,12]. PGAs treatment inhibited *F. solani* growth by destroying the mitochondrial structure [6]. Apoptosis is a programmed cell death (PCD) process, one of the highly conserved cellular eukaryotic suicide programs triggered by external or intrinsic cell stimulation [13]. Our previous study revealed that PGAs treatment could play an antifungal role by inhibiting *F. solani* respiration and energy metabolism [14]. In addition, PGAs treatment had a considerable disruptive effect on cell membranes. Previous studies have indicated that the destruction of cell membrane structure is accompanied by the occurrence of oxidative stress in vivo, resulting in an impact on cell growth [15,16]. However, it is not clear whether PGAs could inhibit *F. solani* by inducing oxidative stress to induce cell apoptosis. Therefore, in this study, *F. solani* was used as the test strain, the induction effect of PGAs on cell apoptosis of *F. solani* was investigated, and the influence of PGAs on the reactive oxygen species (ROS) level of *F. solani* was clarified, with a view to systematically revealing the induction mechanism of PGAs on cell apoptosis of *F. solani* and to provide some reference for effective prevention and treatment of wolfberry root rot.

## 2. Materials and Methods

### 2.1. Test Strain

*Fusarium solani* was isolated from the diseased plants of wolfberry root rot and stored in the Forest Protection Laboratory of Gansu Agricultural University after pathogenicity determination. Before use, it was activated on a potato dextrose agar (PDA) plate and stored at 4 °C for later use.

### 2.2. Extraction of Potato Glycoside Alkaloids (PGAs)

Fresh potatoes were washed and left to bask in the sunlight. Then, the greenish potato skins and sprouts were collected, dried, and crushed. Finally, they were sieved through a 150-mesh sieve and stored for later use.

The extraction of PGAs is based on the method of Zhang et al. [6]. The final residue was dissolved with 20 mL hot ethanol to obtain the extraction mother liquor with a concentration of 197.89 mg·mL^−1^ PGAs.

### 2.3. Determination of the Concentration for 50% of Maximal Effect (EC_50_)

EC_50_ was determined by the mycelium growth rate method. PDA medium with PGAs concentrations of 1.0, 1.2, 1.4, 1.6, 1.8, 2.0, 2.2 mg/mL were prepared. *F. solani* were punched with a sterile 5 mm diameter punch and inoculated on the prepared plate. Sterile water was utilized as the control, with three replicates for each treatment, and cultured in the darkness at 25 °C for 9 days. The virulence regression equation and EC_50_ were calculated.

### 2.4. Preparation of Mycelium

The sterilized cellophane with the same area was spread on the PDA medium containing EC_50_ PGAs, and 5 mm *F. solani* cake was inoculated in the medium, and incubated at 25 °C. The mycelium was collected at 3, 5, 7, and 9 d, respectively. It was frozen in liquid nitrogen after weighing 0.1 g. Finally, they were stored in a −80 °C ultra-low-temperature freezer for reserve.

### 2.5. Determination of Antifungal Effect

#### 2.5.1. Determination of Colony Diameter and Biomass

The obtained concentration of EC_50_ PGAs was added to the PDA medium and shaken well to make the plate, and 5 mm *F. solani* was inoculated in the PDA medium. Colony morphology was observed on different days, and the diameter of the colonies was measured by the criss-cross method.

The mycelium was collected on different days and weighed to obtain the biomass.

#### 2.5.2. Determination of Sporulation and Spore Germination Rate

A total of 10 mL sterile water was added to the plates inoculated with *F. solani* after 3, 5, 7, and 9 d, respectively, and the spreader gently scraped off the spores from the PDA plates. The number of spores was counted by a hemocytometer.

Take 20 μL of *F. solani* spore suspensions (1 × 10^6^ spores/mL) and culture it in the center of the PDA medium containing EC_50_ PGAs. A total of 150–200 spores were examined microscopically at 3, 6, 9, and 12 h of incubation. When the germination tube length is greater than half the spore length, it is recognized as spore germination and the spore germination rate is calculated.

### 2.6. Determination of Metacaspase Enzyme Activity

Caspase-3 activity was measured as metacaspase enzyme activity using the assay kit from Beijing Solarbio Science & Technology Co., Ltd., Beijing, China. When the substrate is saturated, the amount of enzyme that can shear 1 nmol of pNA substrate to produce 1 nmol of free pNA within 1 h at 37 °C is an enzyme activity unit U, and the unit was denoted as U/mg prot.

### 2.7. Propidium Iodide (PI) Stain

PI staining was performed using the method of Yang et al. [17]. An amount of 1 mL of spore suspension *F. solani* (1 × 10^6^ spores/mL) was placed in a 2 mL sterile centrifuge tube, centrifuged at 1000 r/min for 2 min, and the supernatant was discarded. Then, 1 mL of PGAs with EC_50_ was added to the precipitate and incubated at 28 °C for 4 h, centrifuged, and the supernatant was discarded. Next, 1 mL of phosphate buffer was added to the spore precipitate, which was allowed to stand for 30 min at room temperature in the dark, centrifuged, and the supernatant was discarded. Then, 1 mL PBS and 10 μL PI (1 mg/mL) were added to the spore precipitation and left for 20 min in the dark, then centrifuged, and the supernatant was discarded. Finally, 1 mL PBS was added for washing, and after washing three times, the membrane integrity of *F. solani* was observed under a fluorescence microscope and photographed.

### 2.8. Determination of ROS Levels

#### 2.8.1. Determination of NADH Oxidase (NOX) and Superoxide Dismutase (SOD) Activities

NOX activity was determined by using the assay kit from Nanjing Jiancheng Bioengineering Institute, Nanjing, China. An enzyme viability unit U was defined as a 0.01 change in A_600_ per minute per gram of tissue per milliliter of the reaction system, and the unit was denoted as U/g FW. SOD activity was determined by using the assay kit from Beijing Solarbio Science & Technology Co., Ltd., Beijing, China. The amount of enzyme corresponding to 50% SOD inhibition was used as a unit of SOD activity U, denoted as U/g FW.

#### 2.8.2. Determination of Superoxide Anion (O_2_^−^) Production Rate and Hydrogen Peroxide (H_2_O_2_) Content

The production rate of O_2_^−^ was determined by using the assay kit from Nanjing Jiancheng Bioengineering Institute, Nanjing, China. The value of the change in the number of superoxide anion radicals inhibited per gram of sample protein reacted at 37 °C for 40 min equivalent to that inhibited by 1 milligram of vitamin C was taken as one unit of viability U, denoted as U/g prot. The H_2_O_2_ content was determined by using the assay kit from Nanjing Jiancheng Bioengineering Institute, Nanjing, China, and the unit was denoted as mmol/g prot.

#### 2.8.3. Measurement of Intracellular ROS Staining and Fluorescence Intensity

Intracellular ROS staining was performed using 2,7-dichlorodihydro-fluorescein diacetate (DCFH-DA) according to the method of Sun et al. [18]. The level of ROS was observed and photographed under a fluorescence microscope. The fluorescence intensity was measured using Image J software (version 1.52) according to the method of Wang et al. [19].

### 2.9. Determination of Antioxidant Capacity

#### 2.9.1. Determination of Antioxidant Enzyme Activity

The activities of dehydroascorbate reductase (DHAR) and monodehydroascorbate reductase (MDHAR) were determined according to the methods of Ma et al. [20] with some modifications. ΔOD_290_/min/g was defined as an enzyme activity unit of DHAR. ΔOD_340_/min/g was defined as an enzyme activity unit of MDHAR.

Ascorbate peroxidase (APX) activity was determined by using the assay kit from Beijing Solarbio Science & Technology Co., Ltd., Beijing, China. The oxidation of 1 μmol ascorbic acid (AsA) per minute per gram of sample is one enzyme activity unit U, denoted as U/g FW. Glutathione reductase (GR) activity was determined by using the assay kit from Beijing Solarbio Science & Technology Co., Ltd., Beijing, China. The catalytic oxidation of 1 μmol of NADPH per minute per gram of sample was taken as one unit of enzyme activity U, denoted as U/g FW. Catalase (CAT) activity was determined by using the assay kit from Beijing Solarbio Science & Technology Co., Ltd., Beijing, China. The degradation of 1 μmol H_2_O_2_ catalyzed per minute per gram of sample in the reaction system was defined as an enzyme activity unit U, denoted as U/g FW. Thioredoxin peroxidase (TPX) activity was determined by using the assay kit from Nanjing Jiancheng Bioengineering Institute, Nanjing, China. The catalysis of 1 μmol H_2_O_2_ per minute per gram of sample is one unit of enzyme activity U, denoted as U/g FW.

#### 2.9.2. Determination of Antioxidant Substance Content

The determination of AsA and dehydroascorbic acid (DHA) content was modified according to the method of Turcsanyi et al. [21]. An amount of 0.1 g of frozen mycelium was taken, and 1 mL of 100 mM hydrochloric acid was added to the pre-cooled mycelium, then ground into homogenate in the ice bath and centrifuged at 4 °C, 7800 × g for 10 min. The crude enzyme solution was collected and used to determine the contents of AsA and DHA.

The NADPH content was determined by using the assay kit from Nanjing Jiancheng Bioengineering Institute, Nanjing, China, and the unit was denoted as nmol/g FW. The reduced glutathione (GSH) and oxidized glutathione (GSSG) contents were determined by using the assay kit from Beijing Solarbio Science & Technology Co., Ltd., Beijing, China, and the units were denoted as μg/g FW.

#### 2.9.3. Determination of Total Antioxidant Capacity (T-AOC) and Hydroxyl Radical (·OH) Scavenging Capacity

T-AOC was determined by using the assay kit from Beijing Solarbio Science & Technology Co., Ltd., Beijing, China, and the unit was denoted as μmol/mg prot. The scavenging ability of ·OH was determined by using the Nanjing Jiancheng Bioengineering Institute, Nanjing, China. The concentration of H_2_O_2_ in the reaction system was reduced by 1 mM at 37 °C for 1 min per milligram of sample protein as a hydroxyl radical inhibition unit and denoted as U/mg prot.

### 2.10. Statistical Analysis

Excel 2021 was used for data collation and statistics; SPSS 22.0 for significance test and ANOVA; Duncan’s method was used for multiple comparisons analysis and significance at *p* < 0.05; Origin 2022 was used for plotting. Image J software (version 1.52) was used for fluorescence intensity measurements.

## 3. Results

### 3.1. Determination of the Concentration for 50% of Maximal Effect (EC_50_) Values

The virulence regression equation is generally used to evaluate the bioactivity of antifungals and calculate their EC_50_ value. Potato glycoside alkaloids (PGAs) extract at different concentrations (1.0, 1.2, 1.4, 1.6, 1.8, 2.0, 2.2 mg/mL) had certain inhibitory effects on *Fusarium solani*, and colony diameter gradually decreased with the increase of PGAs treatment concentration (Figure 1). The inhibition rate of PGAs on *F. solani* increased with the increase of concentration, showing a certain degree of dose dependence. The inhibition reached a maximum of 100.00%, at a concentration of 2.2 mg/mL. The toxicity regression equation was y = 0.3171x − 0.0886 and the correlation coefficient (R^2^) is 0.9508, indicating that the variables in the regression equation were closely related linearly and the test was feasible. The EC_50_ was 1.85 mg/mL. Subsequent trial concentrations were performed using EC_50_.

### 3.2. The Changes of Colony Diameter, Biomass, Sporulation, and Spore Germination Rate of F. solani After PGAs Treatment

Colony diameter and biomass could directly reflect the growth of mycelium. The colony of the control was growing well. PGAs treatment significantly decreased the colony diameter compared with the control, which was 74.8% and 48.25% lower than the control after 5 d and 9 d (Figure 2A,B). The biomass of the control and the treatment increased with the extension of culture time. PGAs treatment decreased the biomass of *F. solani* compared with the control, which was 70.00% lower than the control after 5 d of culture (*p* < 0.05) (Figure 2C).

The spore germination rate and sporulation are essential for evaluating fungi’s reproductive and spore survival abilities. The sporulation of the control and the treatment increased with the increase of culture days. Compared with the control, PGAs treatment significantly decreased sporulation, which was 92.87% lower than the control after 9 d of culture (*p* < 0.05) (Figure 2D). The control basically germinated at 12 h, and the germination rate was 94.79%. PGAs treatment decreased the spore germination rate, and the increase in germination rate was negligible. It was 84.95% significantly lower than the control after 12 h of culture (Figure 2E). Therefore, PGAs treatment significantly decreased the colony diameter and biomass of *F. solani*, delayed spore germination, and inhibited the sporulation.

### 3.3. Changes of Metacaspase Activity

Caspase-3 is the most essential terminal protease in the process of apoptosis, and its activity is generally used to measure the degree of apoptosis. The activity of caspase-3 in the control remained unchanged, while the activity of caspase-3 in the treatment was significantly increased with the increase of culture time and reached the peak value after 9 d of culture, which was 56.47% higher than the control (*p* < 0.05) (Figure 3). These results verified that PGAs treatment could increase the activity of metacaspase in *F. solani*.

### 3.4. Propidium Iodide (PI) Staining

PI is a commonly used cytosolic fluorescent dye that can be embedded between bases to achieve binding to DNA. The results of PI staining showed that the cell membranes of control spores were intact with no obvious red fluorescence, while the cell membranes of almost all spores that were PGAs-treated were disrupted, emitting strong fluorescence (Figure 4). This revealed that PGAs treatment induced the apoptosis of *F. solani* cells, and may be in the middle and late stages of apoptosis.

### 3.5. Effect of PGAs on the Level of F. solani Reactive Oxygen Species (ROS)

#### 3.5.1. Effects of PGAs on the Activities of NADH Oxidase (NOX) and Superoxide Dismutase (SOD) and the Contents of Superoxide Anion (O_2_^−^) and Hydrogen Peroxide (H_2_O_2_) in *F. solani*

NOX catalyzes the oxidation of NADH by reducing the molecule O_2_ to H_2_O_2_ or H_2_O and is responsible for most ROS production. NOX activity in the control decreased slowly with the increase of culture time. PGAs treatment significantly increased NOX activity compared with the control, which was 1.49 times higher than the control after 3 d of culture (*p* < 0.05) (Figure 5A). SOD is an antioxidant metalloenzyme in living organisms, which can catalyze the disproportionation of O_2_^−^ radicals to produce O_2_ and H_2_O_2_. SOD activity in the treatment and the control showed a downward trend with the increase of culture time. PGAs treatment increased SOD activity compared with the control, which was 1.30 times higher than the control after 5 d of culture (Figure 5B).

O_2_^−^ is an active oxygen species, which is the product of oxygen molecule reduction by a single electron in cells and has a momentous part in cell redox metabolism. The O_2_^−^ production rate of the control increased first and then decreased with the increase of culture time, while the O_2_^−^ production rate of the treatment showed a gradual upward trend. PGAs treatment increased the O_2_^−^ production rate in the early stage compared with the control, which was 37.95% higher than the control after 5 d of culture (*p* < 0.05) (Figure 5C). H_2_O_2_ is a signal molecule that transmits information about ROS, disease resistance, and defense reactions. An increase in its content is easy to cause membrane lipid peroxidation, thus activating various defense reactions in the body. The content of H_2_O_2_ in the control showed a rising trend with the increase of culture time, while the treatment showed a trend of increasing first and then decreasing. PGAs treatment significantly increased the H_2_O_2_ content in the early stage compared with the control and was 1.77 times higher than the control after 5 d of culture (*p* < 0.05) (Figure 5D). The observed trends indicate that PGAs treatment induced the accumulation of ROS in *F. solani* in the early stage.

#### 3.5.2. *F. solani* ROS Staining

The results of DCFH-DA staining showed that only a few spores in the control emitted green fluorescence, and the fluorescence intensity was weak, while the spores emitted dense and bright fluorescence after PGAs treatment, which was 1.69 times significantly higher than the control (Figure 6A,B). The outcomes revealed underscore the notion that PGAs induced ROS accumulation in *F. solani*.

### 3.6. Effect of PGAs on the Activity of Antioxidant Enzymes of F. solani

Catalase (CAT) is one of the critical enzymes of the antioxidant defense system, which scavenges H_2_O_2_ and avoids oxidative stress. During the period of mycelium culture, the CAT activity of the control did not change considerably. Compared with the control, PGAs treatment decreased CAT activity, which was 87.99% lower than the control after 9 d of culture (*p* < 0.05) (Figure 7A). Thioredoxin peroxidase (TPX) is a vital enzyme molecule that plays a vital role in cell redox regulation. The TPX activity of the control increased first and then decreased during the culture period. PGAs treatment decreased TPX activity compared with the control, which was 73.81% lower than the control after 7 d of culture (*p* < 0.05) (Figure 7B). Ascorbate peroxidase (APX) is an important antioxidant enzyme for scavenging ROS and one of the key enzymes in ascorbic acid metabolism. During the mycelium culture period, the APX activity of the control was on the rise, while the treatment showed a trend of increasing first and then decreasing. PGAs treatment significantly decreased APX activity at the later stage compared with the control, which was 87.31% lower than the control after 7 d of culture (*p* < 0.05) (Figure 7C). Glutathione reductase (GR) is the main flavinase that maintains cells’ reduced glutathione (GSH) content. The GR activity of the control and the treatment showed a trend of decreasing first, then increasing, and then decreasing with the increase of culture time. PGAs treatment decreased GR activity compared with the control, which was 30.92% lower than the control after 7 d of culture (*p* < 0.05) (Figure 7D). Dehydroascorbate reductase (DHAR) catalyzes the GSH-dependent reduction of dehydroascorbic acid (DHA) and plays a direct role in the regeneration of AsA. Compared with the control, PGAs treatment decreased DHAR activity, which was 75.80% lower than the control after 9 d of culture (*p* < 0.05) (Figure 7E). Monodehydroascorbate reductase (MDHAR) can catalyze ascorbic acid (AsA) reduction and be instrumental in maintaining the dynamic balance of intracellular AsA. Compared with the control, PGAs treatment significantly increased MDHAR activity, and it was 1.01 times higher than the control after 5 d of culture (*p* < 0.05) (Figure 7F). The analysis yields the inference that PGAs treatment overall decreased the activities of antioxidant enzymes and created an oxidative stress environment.

### 3.7. Effects of PGAs on the Content of Antioxidant Substances of F. solani

AsA can react with free radicals to inactivate or neutralize them, thus acting as an antioxidant. The AsA content in the control decreased first and then increased with the increase of culture time, while the AsA content in the treatment increased first and then decreased. Compared with the control, PGAs treatment increased AsA content in the early stage, which was 56.98% higher than the control after 5 d of culture (*p* < 0.05) (Figure 8A). DHA acts as an antioxidant; it can scavenge free radicals and reduce oxidative damage. The DHA content of the control and the treatment showed a downward trend with the increase of culture time. PGAs treatment significantly increased DHA content in the early stage compared with the control, which was 69.39% higher than the control after 3 d of culture (*p* < 0.05) (Figure 8B). GSH is a major intracellular antioxidant, redox, and cell signaling regulator that reduces H_2_O_2_ and scavenges ROS and nitrogen-containing free radicals to protect cells from oxidative stress damage. The reduction-to-oxidation ratio (GSH/GSSG) is the main dynamic index of cell redox status. PGAs treatment significantly decreased GSH content compared with the control, which was 38.32% lower than the control after 5 d of culture (*p* < 0.05) (Figure 8C). The oxidized glutathione (GSSG) content of the control and the treatment showed a downward trend with the increase in culture time. PGAs treatment decreased the GSSG content in the early stage compared with the control, which was 32.16% lower than the control after 3 d of culture (*p* < 0.05) (Figure 8D). The ratio of GSH/GSSG in the control increased first and then decreased with the increase of culture time. PGAs treatment increased the ratio of GSH/GSSG compared with the control, showing an overall upward trend. It was 48.28% higher than the control after 9 d of culture (*p* < 0.05) (Figure 8E). One of the most critical roles of NADPH in organisms is as an electron supplier for redox reactions. The NADPH content of the control showed a trend of decreasing first and then increasing during the culture period, while the treatment showed a trend of increasing first and then deducing. PGAs treatment significantly decreased NADPH content compared with the control, which was 91.86% lower than the control after 3 d of culture (*p* < 0.05) (Figure 8F). The aforementioned results suggest that PGAs treatment induced dysregulation of the intracellular redox balance in *F. solani*.

### 3.8. Effect of PGAs on the Antioxidant Capacity of F. solani

Total antioxidant capacity (T-AOC) is one of the most critical indicators of an organism’s resistance to oxidative damage and maintenance of a healthy state. During the culture period, the T-AOC of the control showed an upward trend, while the T-AOC of the treatment remained basically unchanged. PGAs treatment significantly decreased the T-AOC of *F. solani* compared with the control, which was 74.54% lower than the control after 9 d of culture (*p* < 0.05) (Figure 9A).

Hydroxyl radical (·OH) is a highly reactive free radical with oxidizing solid power. The ability of the control and the treatment to inhibit ·OH showed a rising trend with the increase of culture time. Still, the increase in the treatment was not significant and gradually tended to be gentle. PGAs treatment decreased the ability to inhibit ·OH compared with the control, which was 56.77% lower than the control after 9 d of culture (*p* < 0.05) (Figure 9B).

## 4. Discussion

Potato glycoside alkaloids (PGAs) have extensive antifungal activity. The present study has demonstrated that PGAs considerably inhibited the colony diameter, biomass, sporulation, and spore germination rate of *Fusarium solani*. It has been reported that PGAs could inhibit the mycelium growth and spore germination of *Curvularia trifolii* of strawberry [22]. This resonates with the outcomes observed in this research, indicating that PGAs could inhibit the growth and development of *F. solani*.

Apoptosis is an irreversible biological phenomenon that exists in the organism itself, and it is instrumental in the growth and development of the organism and the process of reproduction and evolution. In fungi, apoptosis is known as apoptosis-like programmed cell death (PCD), which was first carried out on yeast, and then apoptosis was also found in the filamentous fungus *Mucor racemosus* [23,24]. It has been reported that there are two apoptotic signaling pathways in fungi induced by antifungal agents, including the metacaspase-dependent pathway and the metacaspase-independent pathway [25]. Lipopeptide C_17_ Fengycin B induces early apoptosis of *F. oxysporum* cells in a metacaspase-dependent manner [26]. After treatment of *Penicillium italicum* with trans-cinnamaldehyde, apoptosis was induced, and caspase-3 activity was increased, while the control remained unchanged [27]. Our research has elucidated that PGAs treatment significantly increased the caspase-3 activity of *F. solani*, and rose with the increase of treatment time, indicating that PGAs can induce apoptosis of *F. solani* through the metacaspase-dependent pathway. The weakening of cell vitality accompanies apoptosis. Propidium iodide (PI) is a nucleic acid dye that detects cell membrane integrity and is used to detect cell viability [28,29]. The PI staining outcomes reveal that the fluorescence intensity of spores in the PGAs treatment was much higher than that in the control, indicating that PGAs destroyed the integrity of the *F. solani* cell membrane, induced *F. solani* apoptosis, and decreased cell viability. This result was consistent with the increase of PI fluorescence intensity in the process of *F. graminearum* cell apoptosis induced by 2-hydroxy-4-methoxybenzaldehyde (HMB) treatment by Li et al. [30].

The accumulation of intracellular reactive oxygen species (ROS) is usually associated with the onset of apoptosis. Superoxide anion (O_2_^−^) and hydrogen peroxide (H_2_O_2_) are essential components of ROS. The NADH oxidase (NOX) family is considered the primary source of ROS in eukaryotic cells. O_2_ is converted into O_2_^−^ by catalysis and then rapidly converted into H_2_O_2_ by spontaneous disproportionation or superoxide dismutase (SOD) on the cell wall [31]. The present study has demonstrated that PGAs treatment induced an increase in NOX and SOD activities. This was consistent with the result that lipopeptide C_17_ Fengycin B significantly increased the SOD activity of *F. oxysporum* in Deng et al. [26]. Therefore, we measured the O_2_^−^ production rate and H_2_O_2_ content, and the results revealed that PGAs treatment increased significantly the O_2_^−^ and H_2_O_2_ of *F. solani* in the early stage. This is consistent with the results of Zhang et al. Treatment with 30 mM H_2_O_2_ increased the O_2_^−^ generation rate and H_2_O_2_ content of *Alternaria alternata* and, at the same time, increased the activity of NOX [32]. *Arabidopsis thaliana* TCP21 protein penetrates fungal cell walls and membranes without inducing any membrane alterations and inhibits the growth of fungal hyphae such as mycorrhizal fungi by generating intracellular ROS and mitochondrial superoxide, leading to morphological changes and apoptosis [15]. DCFH-DA is a fluorescent probe for the detection of ROS. DCFH-DA itself is non-fluorescent but it can freely cross the cell membrane. Once it enters the cell, it is hydrolyzed by the intracellular esterase to DCFH, and since DCFH cannot pass through the cell membrane, the fluorescent probe accumulates inside the cell [33]. This experiment showed that *F. solani* emitted green fluorescence after PGAs treatment. In contrast, only a few spores in the control emitted green fluorescence, and the fluorescence was weak and sparse, which was consistent with the ROS level. Therefore, we speculated that PGAs treatment increased SOD and NOX activity, ROS burst, and ultimately apoptosis in *F. solani*.

Fungi have a range of defense systems to detoxify ROS [34]. The ascorbate–glutathione (AsA-GSH) cycle is a primary defense system against the harmful effects of ROS in the cytoplasm, peroxisomes, mitochondria, and exosomes [35]. The AsA-GSH cycle is an effective detoxification mechanism against oxidative stress [36]. In this cycle, ascorbic acid (AsA) and H_2_O_2_ react under the catalysis of ascorbate peroxidase (APX) to form H_2_O and produce two molecules of monodehydroascorbic acid (MDHA), which is reduced to AsA under the catalysis of monodehydroascorbate reductase (MDHAR), and MDHA can be further oxidized to dehydroascorbic acid (DHA) [37,38]. This study uncovers that PGAs treatment decreased the activity of APX, CAT, and dehydroascorbate reductase (DHAR) while increasing the activity of MDHAR. This will prevent MDHA from being reduced to AsA in time, resulting in the accumulation of AsA and DHA. Glutathione reductase (GR) is a flavin protein oxidoreductase, which can reduce oxidized glutathione (GSSG) to reduced glutathione (GSH) in the presence of NADPH and is pivotal in the regeneration of AsA and GSH. The higher ratio of GSH to GSSG means more GSH and more DHA can be altered to AsA by DHAR [39]. The outcomes of the research suggest that PGAs treatment considerably decreased the contents of GSH, GSSG, NADPH, and GR activity. In turn, the GSH/GSSG ratio fluctuated significantly. This result was similar to Lee et al.’s [40] treatment of *Monilinia fructicola* with chlorogenic acid and caffeic acid, which disrupted the redox homeostasis in the fungus. Hydroxyl radical (·OH) is a free radical that is highly toxic and harmful to living organisms. It can make the carbohydrates, amino acids, proteins, nucleic acids, and other substances in the tissue oxidation, oxidative damage, and destruction, leading to cell necrosis or mutation [41]. The total antioxidant level comprises various antioxidant substances and antioxidant enzymes. This study showed that PGAs treatment significantly decreased the ability to inhibit ·OH and total antioxidant capacity (T-AOC). The findings are in alignment with those of Chen et al. [42] that T-AOC exhibited a concentration/time-dependent decrease after treatment of *Shigella flexneri* with the essential oil of *Lindera glauca*. The above results showed that PGAs treatment caused an oxidative stress environment, weakened antioxidant capacity, and disordered intracellular redox homeostasis.

## 5. Conclusions

Potato glycoside alkaloids (PGAs) have good antifungal activity and inhibit mycelium growth, biomass accumulation, sporulation, and spore germination of *Fusarium solani*. PGAs treatment increased NADH oxidase (NOX) and superoxide dismutase (SOD) activity, and promoted the accumulation of superoxide anion (O_2_^−^) and hydrogen peroxide (H_2_O_2_) content. At the same time, it inhibited the activity of antioxidant enzymes and the content of antioxidant substances in the ascorbate–glutathione (AsA-GSH) cycle, resulting in total antioxidant capacity, cell redox homeostasis, and cell membrane destruction, and ultimately induced apoptosis of *F. solani* cells. The possible mechanism of PGAs inhibiting *F. solani* is shown in Figure 10. The conclusions drawn from this study could provide new insight into the development and utilization of PGAs and the effective prevention and treatment of wolfberry root rot.

## Figures and Tables

**Figure 1 antioxidants-13-01537-f001:**
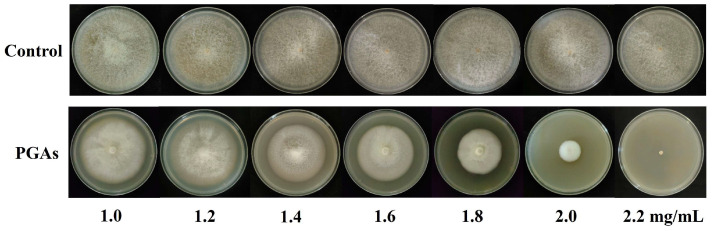
Effects of different concentrations of potato glycoside alkaloids (PGAs) on the growth of *F. solani* colony after 9 d. Note: Control indicates sterile water treatment; PGAs indicates potato glycoside alkaloids treatment (the same as below).

**Figure 2 antioxidants-13-01537-f002:**
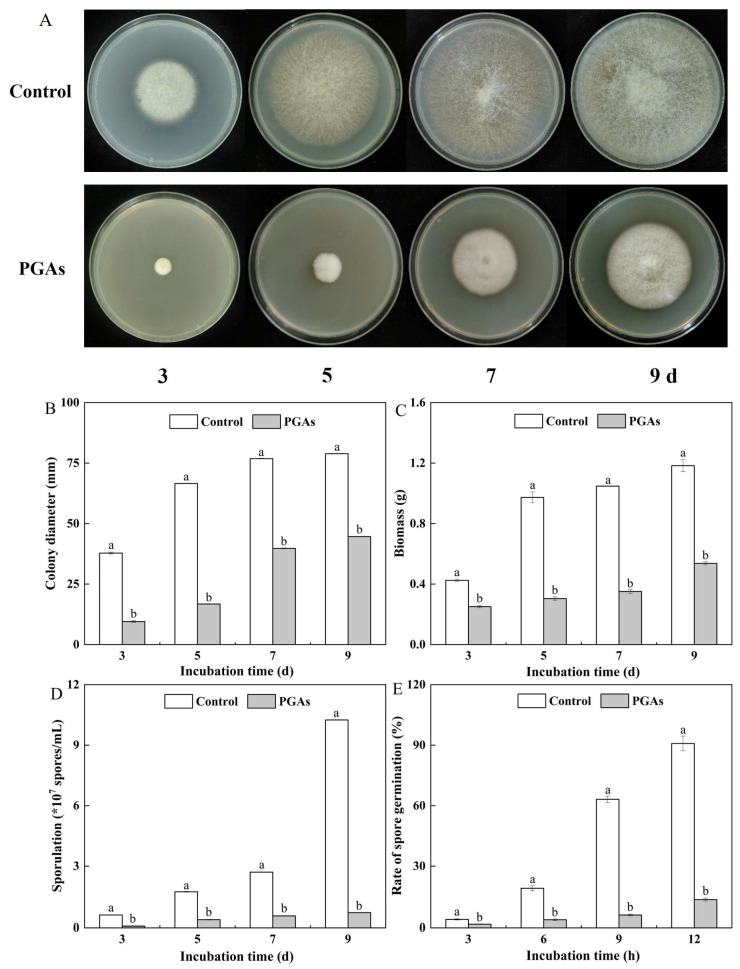
Effects of EC_50_ potato glycoside alkaloids (PGAs) treatment on colony diameter (**A**,**B**), biomass (**C**), sporulation (**D**), and spore germination rate (**E**) of *F. solani*. The vertical line indicates the standard error. Different lowercase letters indicated significant differences between the control and PGAs treatment (*p* < 0.05).

**Figure 3 antioxidants-13-01537-f003:**
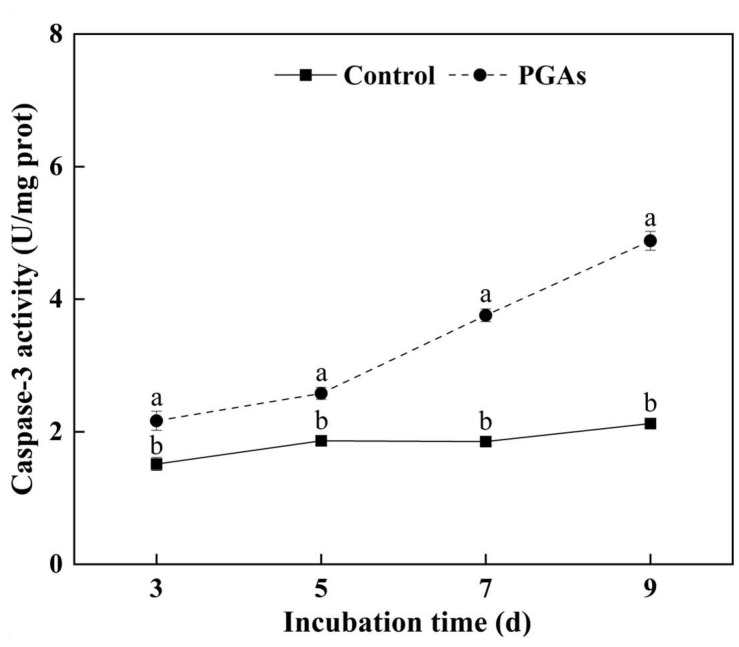
Effect of EC_50_ potato glycoside alkaloids (PGAs) treatment on the caspase-3 activity of *F. solani* at 3, 5, 7, and 9 d. The vertical line indicates the standard error. Different lowercase letters indicated significant differences between the control and PGAs treatment (*p* < 0.05).

**Figure 4 antioxidants-13-01537-f004:**
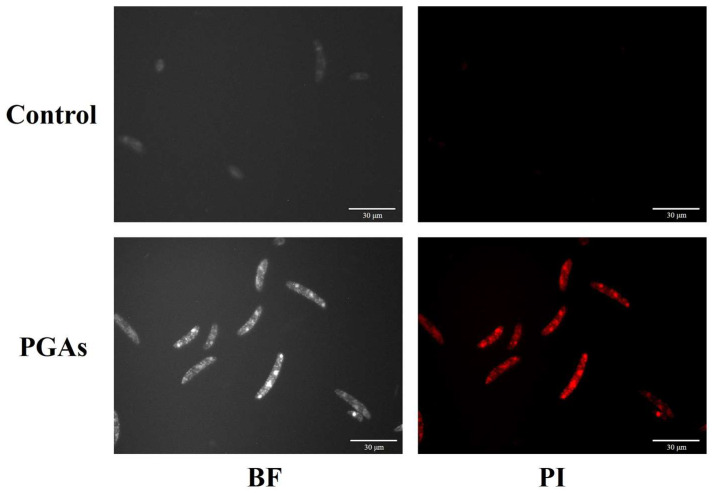
Effect of EC_50_ potato glycoside alkaloids (PGAs) treatment on cell membrane integrity of *F. solani*.

**Figure 5 antioxidants-13-01537-f005:**
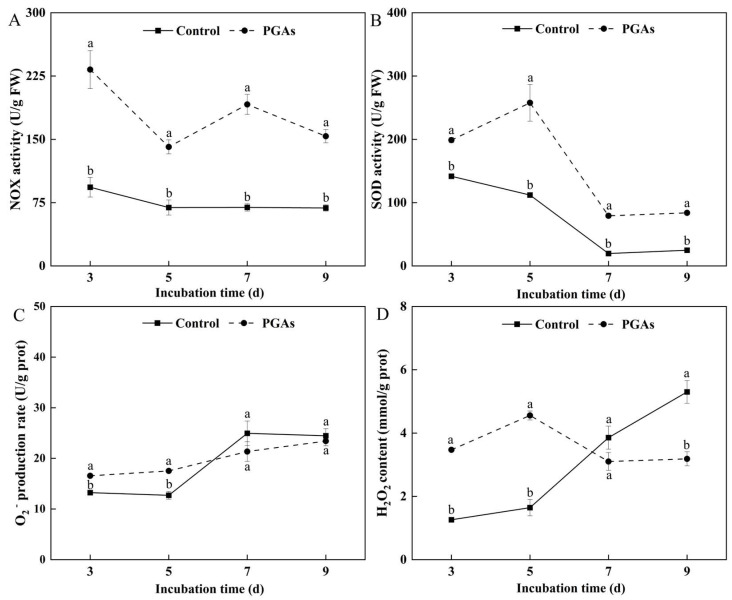
Effects of EC_50_ potato glycoside alkaloids (PGAs) treatment on *F. solani* NADH oxidase (NOX) (**A**), superoxide dismutase (SOD) (**B**), superoxide anion (O_2_^−^) (**C**), and hydrogen peroxide (H_2_O_2_) (**D**). The vertical line indicates the standard error. Different lowercase letters indicated significant differences between the control and PGAs treatment (*p* < 0.05).

**Figure 6 antioxidants-13-01537-f006:**
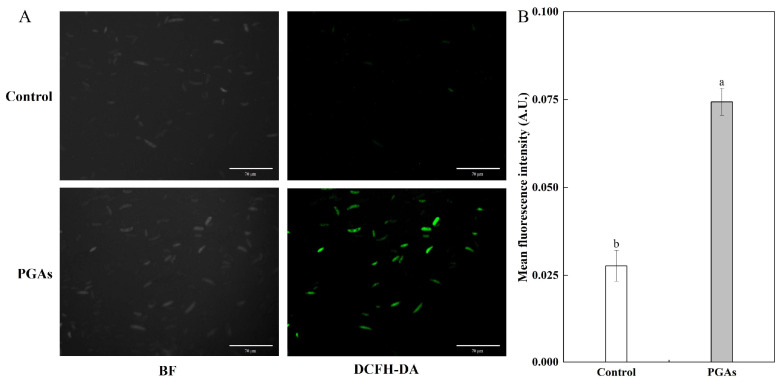
Reactive oxygen species (ROS) staining (**A**) and fluorescence intensity (**B**) of *F. solani*. The vertical line indicates the standard error. Different lowercase letters indicated significant differences between the control and PGAs treatment (*p* < 0.05).

**Figure 7 antioxidants-13-01537-f007:**
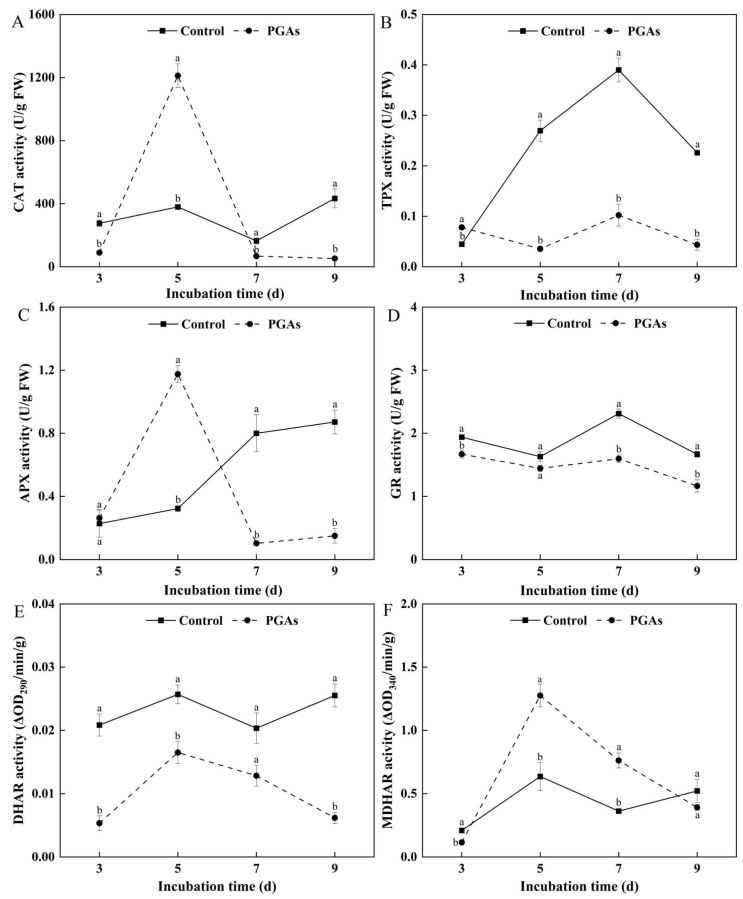
Effects of EC_50_ potato glycoside alkaloids (PGAs) treatment on *F. solani* catalase (CAT) (**A**), thioredoxin peroxidase (TPX) (**B**), ascorbate peroxidase (APX) (**C**), glutathione reductase (GR) (**D**), dehydroascorbate reductase (DHAR) (**E**), and monodehydroascorbate reductase (MDHAR) (**F**). The vertical lines represent standard errors. Different lowercase letters indicated significant differences between the control and PGAs treatment (*p* < 0.05).

**Figure 8 antioxidants-13-01537-f008:**
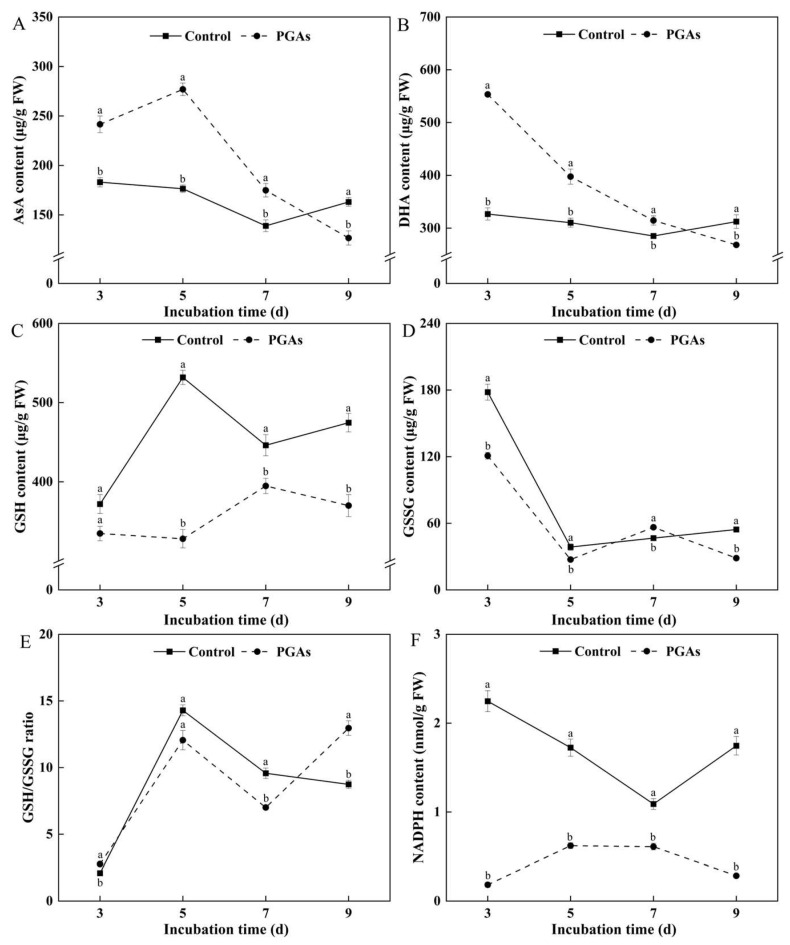
Effects of EC_50_ potato glycoside alkaloids (PGAs) treatment on *F. solani* ascorbic acid (AsA) (**A**), dehydroascorbic acid (DHA) (**B**), reduced glutathione (GSH) (**C**), oxidized glutathione (GSSG) (**D**), GSH/GSSG (**E**), and NADPH (**F**). The vertical lines represent standard errors. Different lowercase letters indicated significant differences between the control and PGAs treatment (*p* < 0.05).

**Figure 9 antioxidants-13-01537-f009:**
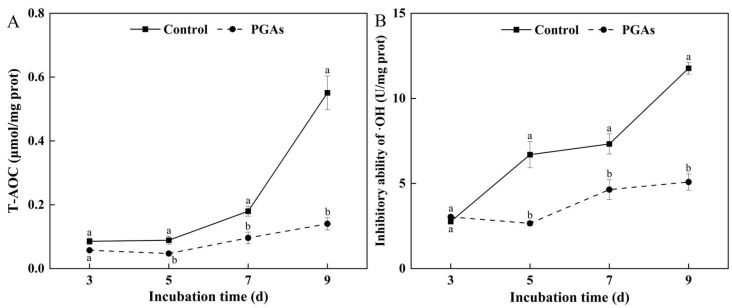
Effects of EC_50_ potato glycoside alkaloids (PGAs) treatment on *F. solani* total antioxidant capacity (T-AOC) (**A**) and inhibition of hydroxyl radical (·OH) (**B**). The vertical lines represent standard errors. Different lowercase letters indicated significant differences between the control and PGAs treatment (*p* < 0.05).

**Figure 10 antioxidants-13-01537-f010:**
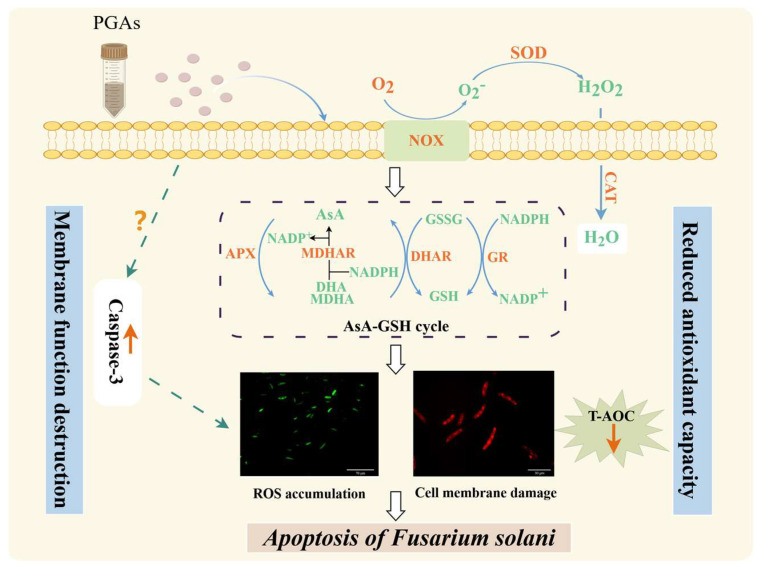
Pattern of antifungal action of potato glycoside alkaloids (PGAs) on *F. solani* (By Figdraw).

## Data Availability

Data are available on request from the authors.

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
