# Peer review of "The Role of Potato Glycoside Alkaloids Mediated Oxidative Stress in Inducing Apoptosis of Wolfberry Root Rot Pathogen Fungi"

_antioxidants, 2024, doi:10.3390/antiox13121537_

Round 1
Reviewer 1 Report
This study investigated the effects of potato glycoside alkaloids (PGA) on inducing cell apoptosis in Fusarium solani and examined how PGA impacts reactive oxygen species levels and antioxidant capacity in F. solani. Overall, it is a well-written manuscript.
Line 75: Consider revising the sentence “The extraction of PGA is on the basis of the method of Zhang et al.” to “The extraction of PGA is based on the method of Zhang et al.”
Author Response
Comments 1: Line 75: Consider revising the sentence “The extraction of PGA is on the basis of the method of Zhang et al.” to “The extraction of PGA is based on the method of Zhang et al.”
Response 1: Thanks for your suggestion. We have revised “The extraction of PGA is on the basis of the method of Zhang et al.” to “The extraction of PGA is based on the method of Zhang et al.” on page 2, paragraph 4, line 78 of the revised manuscript.

Reviewer 2 Report
The manuscript titled. “The role of potato glycoside alkaloids mediating oxidative stress in inducing apoptosis of the wolfberry root rot pathogen” is from the Antioxidants journal area.
This study used potato glycoside alkaloids (PGAs) to investigate their inhibitory effects on F. solani. By analyzing changes in the levels of reactive oxygen species, antioxidant capacity and apoptosis, the role of PGA-induced oxidative stress in inducing apoptosis of F. solani was revealed.
The manuscript presents important results that may contribute to the effective protection of Lycium barbarum roots. It is fairly well edited. The results were obtained in a well-designed study and performed methodologically correctly. The conclusions drawn are correct.
Nevertheless, before publishing the manuscript, I suggest the authors consider the following comments.
1. Abbreviations used for the first time in each chapter should be explained.
2. Abbreviations in figure captions should be clarified.
3. In figures where statistical significance is indicated by asterisks, I suggest replacing the asterisks with letters denoting homogeneous groups.
Nevertheless, before publishing the manuscript, I suggest the authors consider the following comments.
1. Abbreviations used for the first time in each chapter should be explained.
2. Abbreviations in figure captions should be clarified.
3. In figures where statistical significance is indicated by asterisks, I suggest replacing the asterisks with letters denoting homogeneous groups.
Author Response
Comments 1: Abbreviations used for the first time in each chapter should be explained.
Response 1: Thanks for your suggestion. We have added the full name of abbreviations that appear for the first time in each chapter of the revised manuscript.
Comments 2: Abbreviations in figure captions should be clarified.
Response 2: Thanks for your suggestion. We have supplemented the full name of abbreviations in each figure caption on pages 8 to 19, lines 351, 356, 363, 369, 373, 380, 386, 394, 401, and 406 of the revised manuscript.
Comments 3: In figures where statistical significance is indicated by asterisks, I suggest replacing the asterisks with letters denoting homogeneous groups.
Response 3: Thanks for your suggestion. We have replaced the asterisks with lower case letters for homogeneous groups in the figures on pages 8 to 19 of the revised manuscript.

Reviewer 3 Report
The paper entitled “The role of potato glycoside alkaloids mediated oxidative stress 2 in inducing apoptosis of wolfberry root rot pathogen fungi” concerns a pathology of broad interest, and addresses an important aspect of pathogenesis. The text is well written, with sufficient bibliographical references in the introduction and the aim of the work is clear. M&Ms are comprehensive and purpose-appropriate, with sufficient results. Discussion of the results is not speculative.
L17: please use “.” instead of “;”
L20: cell membrane of fungi?
L36: please change in “diseases”
L59: please use italics for in vivo
L69: please add some information about pathovar
L81: not clear what are medicated plates. Maybe do you mean “treated”?
L82: please check the use of medicine-related words in the whole paper: they cannot be used in a plant pathology context.
L83: what a strange acronym for sterile water.
L84: what is the virulence regression equation? Please add details.
L87: “The sterilized cellophane with the same area was spread on the PDA medicated plate” is a sentence not clear not me.
L97: as above, this step is not clear, nor the aim.
L108-109: it is fine to report the company but you cannot report the item code, you need a clear description of the instrument/reagents used.
L114-115: please add some details about the scope of this analysis
L188: just report 100%
L188: this result raises questions, also relating to the choice of different concentrations (L81): why were these concentrations chosen? By what criterion is the maximum concentration 2.2? It is strange that the maximum concentration tested is also the one at which maximum inhibition was obtained. Generally, a test would have foreseen different concentrations, perhaps with some of them exceeding in terms of performance (e.g. it would have been more normal to also test 2.5, 2.7 etc. and note that from 2.2 onwards the inhibition is maximum). It is therefore strange that the highest warhead is also the most efficient.
L327: after how many days?
L331: at which PGA concentration?
L335: please add concentration and days post treatment
L338/340/346/350/354: at which PGA concentration?
Author Response
Comments 1: L17: please use “.” instead of “;”
Response 1: Thank you for pointing this out. We agree with this comment. Therefore, we have modified ";" to “.” on page 1, paragraph 1, line 17 of the revised manuscript.
Comments 2: L20: cell membrane of fungi?
Response 2: Thank you for pointing this out. We agree with this comment. The cell membrane in line 20 is the cell membrane of fungi. Therefore, we have revised “cell membrane” to “cell membrane of fungi” on page 1, paragraph 1, line 20 of the revised manuscript.
Comments 3: L36: please change in “diseases”
Response 3: Thank you for pointing this out. We agree with this comment. Therefore, we have revised "disease" to "diseases" on the page 1, paragraph 2, line 37.
Comments 4: L59: please use italics for in vivo
Response 4: Thank you for pointing this out. We agree with this comment. Therefore, we have put "in vivo" in italics on page 2, paragraph 1, line 60.
Comments 5: L69: please add some information about pathovar
Response 5: Thank you for pointing this out. We agree with this comment. Therefore, we have supplemented the information about the pathogenic fungus on page 2, paragraph 2, line 70 as follows.
Fusarium solani was isolated from the diseased plants of wolfberry root rot and was provided by stored in the Forest Protection Laboratory of Gansu Agricultural University after pathogenicity determination. Before use, it was activated on a potato dextrose agar (PDA) plate and stored at 4°C for later use.
Comments 6: L81: not clear what are medicated plates. Maybe do you mean “treated”?
Response 6: Thank you for pointing this out. We agree with this comment. The medicated plate is a PDA medium with different concentration of PGA. Therefore, we have changed "The PGA mother liquor was added to the PDA medium to formulate a reservoir solution with a final mass concentration of 1.0, 1.2, 1.4, 1.6, 1.8, 2.0, and 2.2 mg/mL to make medicated plates" to " PDA medium with PGA concentrations of 1.0, 1.2, 1.4, 1.6, 1.8, 2.0, 2.2 mg/mL were prepared" on the page 2, paragraph 5, line 82.
Comments 7: L82: please check the use of medicine-related words in the whole paper: they cannot be used in a plant pathology context.
Response 7: Thank you for pointing this out. We agree with this comment. We have checked medicine-related words in the whole paper. Then, we have replaced ”medicine-containing medium” with " prepared plate " on page 2, paragraph 5, line 87 , replaced “medicated plate” with “plate” on page 3, paragraph 1, line 99 and replaced “medicated plate” with "PDA medium containing EC50 PGA" on page 2, paragraph 6, line 91 and page 3, paragraph 4, line 110.
Comments 8: L83: what a strange acronym for sterile water.
Response 8: Thank you for pointing this out. We agree with this comment. Therefore, we have deleted "(CK)" on the page 2, paragraph 5, line 87 and changed "CK" in all the figures to "Control" on pages 8 to 19.
Comments 9: L84: what is the virulence regression equation? Please add details.
Response 9: Thank you for pointing this out. We agree with this comment. Therefore, we have added the description of the virulence regression equation and replaced "EC50 is used as a standard statistic to assess the dose-response relationship" with "Virulence regression equation is generally used to evaluate the bioactivity of antifungals and calculate their EC50 value" on page 4, paragraph 8, line 200.
Comments 10: L87: “The sterilized cellophane with the same area was spread on the PDA medicated plate” is a sentence not clear not me.
Response 10: Thank you for pointing this out. Cellophane is used as an isolation membrane in fungal mycelium culture, which does not affect the normal growth and nutrient absorption of fungi due to its good permeability. In addition, mycelium can be physically separated from the medium, which is convenient for the collection of mycelium and does not introduce impurities. What this means is that we cut cellophane into a circle the same size as the petri dish, sterilize it and lay it flat on the surface of the PDA medium.
Comments 11: L97: as above, this step is not clear, nor the aim.
Response 11: Thank you for pointing this out. We have revised “The sterile cellophane of uniform area was spread on the PDA medicated plate (EC50 PGA). Mycelial mass was measured on different days.” to “The mycelium was collected on different days and weighed to obtain the biomass” on pages 3, paragraph 2, line 102.
Comments 12: L108-109: it is fine to report the company but you cannot report the item code, you need a clear description of the instrument/reagents used.
Response 12: Thank you for pointing this out. According to your suggestion, we have deleted the item code of the kit used. As for the reagent ingredients you proposed, some reagent ingredients of the kit are trade secrets, so they are confidential.
Comments 13: L114-115: please add some details about the scope of this analysis
Response 13: Thank you for pointing this out. We agree with this comment. Therefore, we have added the detailed steps for propidium iodide (PI) stain on page 3, paragraph 6, line 121 to 131, as follows.
1 mL of spore suspension F. solani (1×106 spores/mL) was placed in a 2 mL sterile centrifuge tube, centrifuged at 1000 r/min for 2 min, and the supernatant was discarded. 1 mL of PGA with EC50 was added to the precipitate and incubated at 28°C for 4 h, centrifuged, and the supernatant was discarded. 1 mL of phosphate buffer was added to the spore precipitate, which was allowed to stand for 30 min at room temperature in the dark, centrifuged, and the supernatant was discarded. 1 mL PBS and 10 μL PI (1 mg/mL) were added to the spore precipitation and left for 20 min in the dark, then centrifuged, and the supernatant was discarded. Finally, 1 mL PBS was added for washing, and after washing 3 times, the membrane integrity of F. solani was observed under a fluorescence microscope and photographed.
Comments 14: L188: just report 100%
Response 14: Thank you for pointing this out. With the increase of PGA concentration, the inhibition rate on Fusarium solani also increased. When the PGA concentration reached 2.2 mg/mL or above, the growth of Fusarium solani was completely inhibited.
Comments 15: L188: this result raises questions, also relating to the choice of different concentrations (L81): why were these concentrations chosen? By what criterion is the maximum concentration 2.2? It is strange that the maximum concentration tested is also the one at which maximum inhibition was obtained. Generally, a test would have foreseen different concentrations, perhaps with some of them exceeding in terms of performance (e.g. it would have been more normal to also test 2.5, 2.7 etc. and note that from 2.2 onwards the inhibition is maximum). It is therefore strange that the highest warhead is also the most efficient.
Response 15: Thank you for pointing this out. We also explain this in comment 14. The reason why we set these concentrations is that we have previously conducted pre-experiments and configured media with PGA concentrations of 1.0, 2.0, 4.0, 8.0, and 16.0 mg/mL for screening for the concentration for 50% of maximal effect (EC50). The results showed that Fusarium solani colony hardly grew when the concentration was 2.0 mg/mL, and the mycelial growth was completely inhibited when the concentration was 4.0 mg/mL. Therefore, we set seven concentrations of 1.0, 1.2, 1.4, 1.6, 1.8, 2.0, and 2.2 mg/mL in the concentration range of 1.0-2.2 mg/mL. After 9 d of culture, it was found that there was no mycelium growth on the medium with a concentration of 2.2 mg/mL, and the inhibition rate reached 100%. Then, we calculated the virulence regression equation y = 0.3171x-0.0886 according to the colony diameter after seven concentration treatments, and replaced the inhibition rate (y) by 50% to calculate the EC50 of 1.85 mg/mL.
Comments 16: L327: after how many days?
Response 16: Thank you for pointing this out. We agree with this comment. We screened the EC50 test by measuring the colony diameter after 9 d of culture and calculating the EC50 value. Therefore, we have added the PGA treatment days and changed “Effects of different concentrations of potato glycoside alkaloids (PGA) on the growth of F. solani colony” to “Effects of different concentrations of potato glycoside alkaloids (PGA) on the growth of F. solani colony after 9 d” on page 8, paragraph 1, line 351.
Comments 17: L331: at which PGA concentration?
Response 17: Thank you for pointing this out. We agree with this comment. Since EC50 (1.85mg/mL) was screened out in section 3.1, subsequent tests were conducted with this concentration. Therefore, we have added the concentration of PGA and changed “Effects of potato glycoside alkaloids (PGA) treatment on colony diameter (A and B), biomass (C), sporulation (D) and spore germination rate (E) of F. solani” to “Effects of EC50 potato glycoside alkaloids (PGA) treatment on colony diameter (A and B), biomass (C), sporulation (D) and spore germination rate (E) of F. solani” on page 10, paragraph 1, line 356.
Comments 18: L335: please add concentration and days post treatment
Response 18: Thank you for pointing this out. We agree with this comment. Therefore, we have added the treatment days and concentration and replaced “ Effect of potato glycoside alkaloids (PGA) treatment on the Caspase-3 activity of F. solani ” with “ Effect of EC50 potato glycoside alkaloids (PGA) treatment on the Caspase-3 activity of F. solani at 3, 5, 7, and 9 d” on page 11, paragraph 1, line 363.
Comments 19: L338/340/346/350/354: at which PGA concentration?
Response 19: Thank you for pointing this out. We agree with this comment. Therefore, we have added the concentration of PGA (EC50) on page 12, paragraph 1, line 369, page 13, paragraph 1, line 373, page 16, paragraph 1, line 386, page 18, paragraph 1, line 394, and page 19, paragraph 1, line 401.
